# Improved 3D Pavement Texture Reconstruction Method Based on Interference Fringe via Optimizing the Post-Processing Method

**DOI:** 10.3390/s23104660

**Published:** 2023-05-11

**Authors:** Chu Chu, Ya Wei, Haipeng Wang

**Affiliations:** Key Laboratory of Civil Engineering Safety and Durability, Ministry of Education, Department of Civil Engineering, Tsinghua University, Beijing 100084, China

**Keywords:** 3D pavement texture reconstruction, interference fringe, pavement surfaces, unequal incident angles

## Abstract

The surface quality of pavement has a significant influence on the driving comfort and the skid resistance performance of roads. The 3D pavement texture measurement provides the basis for engineers to calculate the pavement performance index, such as the international roughness index (IRI), the texture depth (TD), and the rutting depth index (RDI), of different types of pavements. The interference-fringe-based texture measurement is widely used because of its high accuracy and high resolution, by which the 3D texture measurement has excellent accuracy in measuring the texture of workpieces with a diameter of <30 mm. When measuring the engineering products with a larger area (or larger areas), such as pavement surfaces, however, the accuracy is deficient because unequal incident angles due to the beam-divergence angle of the laser beam are ignored during the postprocessing of the measured data. This study aims to improve the accuracy of 3D pavement texture reconstruction based on the interference fringe (3D-PTRIF) by considering the influence of the unequal incident angles during postprocessing. It is found that the improved 3D-PTRIF has better accuracy than the traditional 3D-PTRIF, reducing the reconstruction errors between the measured value and the standard value by 74.51%. In addition, it solves the problem of a reconstructed slant surface, which deviates from the horizontal plane of the original surface. Compared to the traditional post-processing method, for the case of smooth surface, the slope can be decreased by 69.00%; for the case of coarse surface, the slope can be decreased by 15.29%. The results of this study will facilitate accurate quantifying of the pavement performance index by using the interference fringe technique, such as IRI, TD, and RDI.

## 1. Introduction

With higher demand for driving comfort and skid resistance performance regarding pavement, engineers are exploring efficient methods and techniques to acquire the pavement-performance-related parameters, such as the international roughness index (IRI), texture depth (TD), and rutting depth index (RDI), of pavement materials. Such parameters, which help engineers to learn about the skid resistance performance of pavement, are closely related to the characteristics of the pavement texture. Moreover, the texture depth data acquired by the 3D pavement texture measurement can provide the basis for the calculation of pavement-performance-related parameters. Thus, the 3D pavement texture measurement with high accuracy has been investigated.

The 3D-PTRIF, which belongs to interferometry, has excellent accuracy and resolution in acquiring the texture depth data of the object surface [1]. In 1983, Takeda and Mutoh verified that profilometry based on interference fringe is capable of measuring 3D object shapes [2]. In the 1990s, some scholars conducted research on dual-wavelength heterodyne profilometry based on interference fringe to measure uneven object surfaces [3,4,5,6,7]. In 1991, Bone solved the problem of unwrapping in the extraction process for 3D data of interferometry [8] such that the extracted phase data are not limited within [−π, π] and the continuous phase is obtained. In 2001, Su and Chen concluded that Fourier transform is the key process of extracting 3D information from the interference fringe image [9]. This technology can be used to recognize the texture of engineering materials, which facilitates the analysis of surface quality. In order to promote the development of this technology, Su conducted several studies on interferometry [9,10,11,12] in the last decade. In 2012, Lally et al. developed an interferometry to measure the profiles of aggregates. Fourier transform is used to reconstruct the aggregate surface. Its resolution can reach 0.01 mm [13,14]. In 2013, Duan investigated the whole process of 3D reconstruction based on interference fringe [15], and the post-processing method was improved to measure the 3D profiles of workpieces. In 2019, Chu et al. developed a pavement profilometry based on interference fringe that can reconstruct the texture of pavement surfaces. It used the traditional post-processing method and promoted the development of pavement profilometry to an extent [16]. The research on 3D reconstruction of pavement helps engineers to explore the performance characteristics of asphalt mixtures [17]. In 2021, Zhang et al. developed an acceleration algorithm for 3D profilometry based on interference fringe that optimizes the reconstruction effect of interferometry [18]. However, measurement errors inevitably exist in the process of postprocessing by using the interference fringe method because the unequal incident angles are ignored during the process of postprocessing. The unequal incident angles are attributed to the beam-divergence angle of the laser beam, which results in a different real light path of each point on the fringe pattern. Normally, the laser incident angle of each point on the fringe pattern is regarded the same as the laser incident angle of the light axis [16,19,20] in the process of postprocessing because the difference in each light incident angle has negligible influence on the measuring results of workpieces with the diameter of <30 mm. If measuring engineering products with larger sizes, such as pavement surfaces, however, the traditional post-processing method will cause the reconstructed surface to slant due to the different laser incident angle. In order to make the 3D reconstruction more accurate, optimization for the post-processing method is indispensable [21]. The innovation of this research is to solve this problem of slant-reconstructed surface by optimizing the post-processing method of 3D-PTRIF to make it possible to measure a pavement surface with larger area more accurately. This study focuses on the post-processing method of 3D-PTRIF. Its target is to improve the accuracy of 3D-PTRIF by taking into account the influence of unequal incident angles during the process of postprocessing. This paper is organized as follows: Section 2 introduces the implementation of 3D-PTRIF. Section 3 explains the principles of 3D-PTRIF and how the unequal incident angles affect the measurement of pavement texture depth. Section 4 illustrates the optimization of the post-processing method and elaborates on the detailed process of optimizing the post-processing method. Section 5 validates the effect of the optimized post-processing method on the reconstructed smooth surface and the coarse surface. Section 6 provides the conclusion.

## 2. The System of 3D-PTRIF

### 2.1. The Implementation of 3D-PTRIF

The 3D-PTRIF, a system with quasi-static measurement, is designed for measuring the texture depth of a pavement surface and reconstructing its 3D view. The system of 3D-PTRIF is displayed in Figure 1.

The scanning system of 3D-PTRIF moves to the pavement surface being measured, and then it stops to project the fringe pattern onto the surface being measured and the fringe image is captured by a charge-coupled device (CCD). The interference fringe image can help to quantify the inconspicuous unevenness of the pavement surface. The fringe image captured by the CCD can achieve 2 million pixels. In other words, a fringe image can provide 2 million texture depth data. As a result, when the pavement surface being measured has the area of 100 mm∗100 mm, the resolution of the system can reach 0.07 mm. The system has several advantages, such as fast and efficient scanning mode [1,19]. Most of all, it exploits quasi-static measurement, which makes it more convenient to acquire high accuracy. Subsequently, the fringe image is postprocessed to acquire texture depth data.

The pavement surface shown in Figure 2 has slight undulations.

The interference fringe pattern projected onto this pavement area is shown in Figure 2b.

The corresponding 3D reconstruction map is provided in Figure 2c. As shown in Figure 2c, the 3D profile of the pavement surface is drawn by different colors according to the magnitude of the depth values. Yellow indicates the largest depth value, and dark blue refers to the smallest depth value.

### 2.2. The Operational Process of 3D-PTRIF

The 3D-PTRIF has five main processes:

Process 1. The system of 3D-PTRIF has wheels to move to specific location, that makes the pavement surface being measured right under the CCD.

Process 2. Turn on the laser source and the fringe generator can project the interference fringe pattern onto the pavement surface being measured.

Process 3. The fringe image of pavement surface being measured is captured by CCD and then saved by computer.

Process 4. Post processing is performed on the fringe image. During the process of post processing, the phase information is extracted from the fringe image first. Then, the texture depth data are calculated.

Process 5. According to the texture depth data, the 3D reconstruction map of the pavement surface being measured is acquired.

Postprocessing (Process 4) has two parts. The second part is calculating the texture depth data, which is the key point of this research.

## 3. The Influence of Unequal Laser Incident Angles

Before revealing the influence of unequal laser incident angles, the relationship between the texture depth data and the laser incident angle should be explained.

### 3.1. The Relationship between the Texture Depth Data and the Laser Incident Angle

The principle of obtaining the texture depth data of the surface being measured, which involves the relationship between the texture depth data and the laser incident angle, is illustrated in Figure 3: When two beams of coherent light encounter each other in the space, zebra stripes, which are named interference fringe, will be exhibited on the projected surface. The fringe generator projects an interference fringe pattern onto the pavement surface being measured. The pavement surface is uneven because of its texture and roughness, which results in the deformation of the fringe pattern. The texture depth data are then acquired, according to the fringe image captured by CCD, and postprocessed by the computer. A standard component with a smooth plane is regarded as the reference plane that is set onto the pavement surface being measured.

As illustrated in Figure 3, R is an arbitrary point on the object surface to be measured. D refers to the output end of the fringe generator. C is the output end of the charge-coupled device (CCD). t is the horizontal distance from C to D. AC and BD are the light axis of the CCD and the fringe generator, respectively, which intersect at R. The location of A is $x_{A}$ and the location of B is $x_{B}$. d is the height from the output end of CCD to the reference plane. h(x) refers to the texture depth perpendicular to the direction of X axis. ΔARB is similar to ΔDRC, obtaining that:(1)$$\frac{t}{\left|x_{B}-x_{A}\right|}=\frac{d-\left|h\left(x\right)\right|}{\left|h\left(x\right)\right|}$$
(2)$$\mathrm{AB}=\left|x_{B}-x_{A}\right|=\Delta \varphi \left(x\right)/2\pi f_{0}$$
where Δφ(x) is the fringe phase difference between the uneven surface and the reference plane [6]. $f_{0}$ is the fringe frequency, which is expressed as Equation (3).
(3)$$f_{0}=\frac{2a\cos\emptyset }{\lambda d_{p}}$$
where 2a refers to the fiber core interval, ϕ is the laser incident angle when the fringe pattern is projected onto the reference plane, λ is the laser wavelength, and $d_{p}$ is the length of BD. Two beams of coherent light, which are emitted by two fibers, generate reference fringe. Fiber core interval indicates the interval between the cores of the two fibers.

Substituting Equation (2) into Equation (1), the texture depth |h(x)| is obtained:(4)$$\left|h\left(x\right)\right|=\frac{d\Delta \varphi \left(x\right)}{2\pi f_{0}t+\Delta \varphi \left(x\right)}$$

Substituting Equation (3) into Equation (4), it is obtained that:(5)$$\left|h\left(x\right)\right|=\frac{\lambda d_{p}^{2}\Delta \varphi \left(x\right)\cos\phi }{4a\pi t\cos\phi +\lambda d_{p}\Delta \varphi \left(x\right)}$$

According to Equation (5), the calculated texture depth is related to the laser incident angle.

### 3.2. The Influence of Unequal Incident Angles on the Calculation of Texture Depth

Unequal incident angles have an influence on the calculation of texture depth. As shown in Figure 4, the fringe generator projects the fringe pattern onto the surface being measured: the red area is the fringe pattern and β is the beam-divergence angle. The beam-divergence angle β of the laser beam results in that the real light path of each point, such as $d_{1}$ and $d_{2}$, on the fringe pattern is different. Thus, the laser incident angle $\phi^{\prime }$ of each point is different. Further, $\phi_{1}$ and $\phi_{2}$ are the largest and smallest laser incident angles of the laser beam, respectively.

As Equation (5) indicates, different laser incident angle $\phi^{\prime }$ of each point, induced by the beam-divergence angle β, has an influence on the calculated texture depth. Previously, the influence was ignored because the viewing field was usually small, and the incident angle of each point on the fringe pattern was regarded as the incident angle ϕ of the laser axis. This behavior causes erroneous measuring results, which makes the reconstructed surface slant, as shown in Figure 5: the angle of slope means the angle between the horizontal plane and the trend of the reconstructed surface. The yellow area represents the point cloud with larger depth values and the blue area represents the point cloud with smaller depth values. The larger the depth values are, the darker the color of the point cloud is. As shown in Figure 5, one side of the reconstructed surface is light yellow and the other side is blue. Along the side length of the reconstructed surface, the color changes from yellow to blue, which means the reconstructed surface is slanted.

In summary, the existing problem is that ignoring the difference in laser incident angles will introduce errors to the reconstructed surface. Previously, laser incident angle of each point on the fringe pattern is regarded as the same with the light axis, attributed to that the difference in each laser incident angle has negligible influence on the measuring results of workpieces with the length of <30 mm. If measuring engineering products with larger size, such as pavement surfaces, the postprocessing method has to be optimized. 

## 4. The Optimization for Post-Processing

The acquisition for the texture depth of 3D pavement texture reconstruction based on interference fringe requires post-processing in order to extract 3D information from the fringe image. It is inevitable to introduce errors during the process of postprocessing. However, optimization for post-processing method can reduce errors that are brought in during the process of postprocessing. 

This section introduces the traditional post-processing method first and then elaborates on the optimization for the traditional post-processing method.

### 4.1. The Traditional Post-Processing Method

During the process of postprocessing, the fringe image must be processed first.

A fringe image can be expressed by Equation (6) [22,23,24]:(6)$$g\left(x,y\right)=I_{0}\left(x,y\right)+r\left(x,y\right)\cos[2\pi f_{0}x+\varphi \left(x,y\right)]$$
where g(x, y) is the light intensity of the fringe pattern, $I_{0}\left(x,\;y\right)$ is the background light intensity, r(x, y) is the fringe contrast, φ(x,y) is the fringe phase, and $f_{0}$ is the fringe frequency.

As shown in Figure 4, the light path to the surface being measured is different from that to the reference plane, so the fringe phase of the fringe pattern on the surface being measured is different from that on the reference plane. $\varphi_{1}\left(x,y\right)$ is the fringe phase of the uneven surface being measured; $\varphi_{0}\left(x,y\right)$ is the fringe phase of the reference plane.
(7)$$\Delta \varphi \left(x,y\right)=\varphi_{1}\left(x,y\right)-\varphi_{0}\left(x,y\right)$$

Fourier transform plays a role in extracting Δφ(x,y) from the fringe image [6,7,8], which means the processing on the fringe image is accomplished and then the texture depth data will be calculated.

The depth of arbitrary point on the object surface is worked out according to Equation (5). Then, the texture depth data of point cloud on the object surface can be integrated.

In order to calculate the texture depth data of the surface being measured, a reference plane is required to be compared. All the operations are performed on both fringe images of the surface being measured and the reference plane. Figure 6 shows the traditional post-processing method of acquiring the texture depth data in a detailed manner:

First, the fringe images are converted from time domain to frequency domain by Fourier transform in order to extract the fundamental frequency component, which includes phase information from the frequency domain. Then, a bandpass filter is used to extract the fundamental frequency component from the frequency domain [25]. Afterwards, inversed Fourier transform is performed on the fundamental frequency component in order to transform the fundamental frequency component back into the time domain. Subsequently, the transformed fundamental frequency component $g^{*}\left(x,y\right)$ is obtained. The wrapped phase difference Δφ(x, y) is calculated according to Equation (8).
(8)$$\Delta \varphi \left(x,y\right)=arctan\left(\frac{I_{m}\left(g^{*}\left(x,y\right)\cdot g_{0}^{*}\left(x,y\right)\right)}{R_{m}\left(g^{*}\left(x,y\right)\cdot g_{0}^{*}\left(x,y\right)\right)}\right)$$
where $g^{*}\left(x,y\right)$ is the transformed fundamental frequency component of the surface being measured performed by inversed Fourier transform; $g_{0}{}^{*}\left(x,y\right)$ is the transformed fundamental frequency component of the reference plane performed by inversed Fourier transform; $I_{m}$ is the real part and $R_{m}$ is the imaginary part. 

The wrapped phase difference Δφ(x, y) is wrapped in [−π, π] due to the feature of arctangent function. Subsequently, Δφ(x, y) is unwrapped. At last, the depth of an arbitrary point:(9)$$\left|h\left(x,y\right)\right|=\frac{d\Delta \varphi \left(x,y\right)}{2\pi f_{0}t+\Delta \varphi \left(x,y\right)}$$

The traditional post-processing method of texture measurement based on interference fringe ignores the influence of unequal incident angles (as shown in Equation (9)), which have negligible effect on measuring the texture of workpieces with the diameter of <30 mm. However, when the traditional method measures the engineering products with larger area, such as pavement surfaces, the accuracy is deficient. As a result, the post-processing method is required to be optimized.

### 4.2. The Optimized Post-Processing Method

(1)The reason for optimization

The reason for optimization is that the real laser incident angle of each point on the fringe image, such as $\phi_{1}$ and $\phi_{2}$_,_ was regarded the same with the incident angle ϕ of the laser axis, as shown in Figure 4. Most importantly, the ignoring of unequal laser incident angles causes the reconstructed surface slant that is deviated from the horizontal plane of the original surface. Therefore, the unequal incident angles are taken into account by optimizing the post-processing method for the 3D reconstruction of pavement surface.

As shown in Figure 4, $d_{1}$ and $d_{2}$ are the longest and the shortest distance between the output end of the fringe generator and the certain point on the fringe pattern, respectively.

d can be expressed by $\phi^{\prime }$:(10)$$d_{p}=\frac{d}{\cos\phi^{\prime }}=d\cdot \sec\phi^{\prime }$$

Substituting Equation (10) into Equation (5), we obtain:(11)$$\left|h\left(x\right)\right|=\frac{\lambda d^{2}\Delta \varphi \left(x\right)}{4a\pi t\cos^{2}\phi^{\prime }+\lambda d\Delta \varphi \left(x\right)}$$

(2)The method of optimization

The fringe image, captured by digital camera, is a digital matrix. As shown in Figure 4, each column of the matrix has different $\phi^{\prime }\left(12{}^{\circ}\leq \phi^{\prime }\leq 15{}^{\circ}\right)$ because the laser beam has a divergence angle. When d=1000 mm, the diameter of the fringe image is 91.96 mm [16], and the beam-divergence angle β of the laser beam emitted by the fringe generator:(12)$$\beta =2\arctan(\frac{91.96}{2}/1000)=5{}^{\circ}15^{\prime }54^{\prime \prime }$$

Whatever the variation of *d* is, the beam-divergence angle *β* is constant. Therefore, the real value of $\phi^{\prime }$ is in the range of (ϕ−β/2, ϕ+β/2), as shown in Figure 4. *ϕ* is the incident angle of the laser beam. 

In terms of this system, when d=1000 mm, 1 mm≈15 pixels, so the resolution of a fringe image is 1379×1379 pixels. When d takes other values, 1 mm≈Pi pixels, and then the diameter of a fringe image expressed by pixel number is:(13)$$\Delta x_{pixel}=\frac{2\lambda P_{i}d}{\pi W_{P}}$$
where $P_{i}$ refers to the pixel number contained in the width of 1 mm. $W_{P}$ is the diameter of the fiber core. Therefore, the pixel number of each column is $\frac{2\lambda P_{i}d}{\pi W_{P}}$. The beam-divergence angle of all the columns is β; the beam-divergence angle of each pixel is $\frac{\pi \beta W_{P}}{2\lambda P_{i}d}$. As discussed above, $\phi^{\prime }$ is in the range of (ϕ−β/2, ϕ+β/2), and each column of the matrix is corresponded with a value of $\phi^{\prime }$: (14)$$\phi^{\prime }=\phi -\frac{\beta }{2}+\frac{\pi \beta W_{P}}{2\lambda P_{i}d}j$$
where j represents column j of the matrix.

Substituting Equation (14) into Equation (11), h(:, j), which means the depth values of Column j on a fringe image, is
(15)$$\left|h\left(:,j\right)\right|=\frac{\lambda d^{2}\Delta \varphi \left(:,j\right)}{4a\pi t\cos^{2}(\phi -\frac{\beta }{2}+\frac{\pi \beta W_{P}}{2\lambda P_{i}d}j)+\lambda d\Delta \varphi \left(:,j\right)}$$

In general, the last process of acquiring the texture depth data is shown in Figure 7.

## 5. Validation of the Optimized Post-Processing Method

In order to validate the effect of the optimized post-processing method, a standard component of texture depth, a piece of asphalt pavement, and a piece of cement pavement are tested, respectively. The standard component is the one we used in Section 2. The pavement surfaces both have the area of 100 mm∗100 mm. The standard component of texture depth, which is made of stainless steel, is smooth surface. The pavement surfaces are coarse surfaces. Interference fringe is projected onto these surfaces being measured, and the fringe images are captured. The postprocessing is then performed on the fringe images for 3D reconstruction of these surfaces. The traditional post-processing method and the optimized post-processing method are both performed on the fringe images. The effect of the optimized post-processing method is evaluated by comparing with the traditional post-processing method.

### 5.1. Standard Component of Texture Depth

The standard component of texture depth, which is used to validate the effect of the optimized post-processing method on the smooth surface, has the width of 3 mm and the depth of  2 mm, with the manufacture accuracy of ±0.2 mm, as shown in Figure 8a. Its roughness error is ±0.2 mm. After being scanned by 3D-PTRIF, the corresponding fringe image is acquired, as shown in Figure 8b. The traditional post-processing method and the optimized post-processing method are used to reconstruct the upper surface of the standard component. As shown in Figure 8c, the reconstructed 3D profile is drawn by different colors according to the magnitude of the depth values. The blue area on the reconstructed surface refers to the groove on the standard component. Further, 800 points on the midcourt line of the reconstructed groove, which are acquired by the traditional post-processing method and the optimized post-processing method, respectively, are used to compare the slopes along the reconstructed grooves. The least squares method is used to complete curve fitting with the 800 points on the midcourt line of the reconstructed groove, and the slope along the reconstructed groove is calculated according to the following equation:(16)$$k=\frac{\sum_{i=1}^{800}\left(x_{i}-\bar{x}\right)\left(z_{i}-\bar{z}\right)}{{\left(x_{i}-\bar{x}\right)}^{2}}$$
where *k* indicates the slope along the X axis of the coordinate system, $x_{i}$ refers to the location of pixel *i* along the X axis of the coordinate system, $\bar{x}$ refers to the average value of $x_{i}$, $z_{i}$ is the depth corresponding with $x_{i}$, and $\bar{z}$ is the average value of $z_{i}$.

The slope gradients of the reconstructed grooves are summarized in Table 1.

The texture depth of each point on the reconstructed surface is compared with the known dimensions of the standard texture depth. It is obtained that the largest error is ±0.51 mm with the traditional post-processing method and ±0.13 mm with the proposed post-processing method. The reduction rate of error is 74.51%. As shown in Table 1, the slope of the reconstructed surface is decreased from 0.0016 to −4.9598 × 10^−4^. The decrease in the slope is 69.00%.

### 5.2. Asphalt Pavement Surface

In order to evaluate the optimization effect of the proposed post-processing method, the optimized post-processing method and traditional post-processing method are performed on the fringe image to acquire the texture depth data, respectively. Figure 9a,b are the surfaces reconstructed by the traditional post-processing method and the optimized post-processing method, respectively.

The asphalt pavement surface to be measured, which is used to validate the effect of the optimized post-processing method on the coarse surface, has the area of 100 mm∗100 mm, as shown in Figure 9a. After the interference fringe pattern is projected onto it, the pavement fringe image is captured, as shown in Figure 9b. The measuring height is 1000 mm. Figure 9c,d shows the reconstructed surfaces acquired by the traditional post-processing method and the optimized post-processing method. The reconstructed 3D profile is drawn by different colors according to the magnitude of the depth values.

The slopes of certain contour lines on the reconstructed surfaces are calculated with the least squares method (Equation (16)) in order to compare the slopes of both reconstructed surfaces. In the coordinate system of the reconstructed surface, as shown in Figure 10, each row corresponds with a contour line. The contour lines of y = 10 mm, 30 mm, 50 mm, 70 mm, and 90 mm are selected to evaluate the slope of the reconstructed surface because they are evenly distributed on the reconstructed surface. The slope gradients of the reconstructed surfaces are summarized in Table 2.

As shown in Table 2, the slopes of AB, CD, EF, GH, and IJ on the surface reconstructed by the optimized post-processing method all become smaller than that reconstructed by the traditional post-processing method. The surface slope reconstructed by the optimized post-processing method is decreased by 18.42% compared with the traditional post-processing method.

The traditional post-processing method makes the reconstructed surface slant. On the condition that the slope is constant, the errors will increase when the length of the reconstructed surface increases. The proposed post-processing method solves this problem of slant-reconstructed surface. As shown in Table 2, the slope of the surface reconstructed by the proposed post-processing method is smaller than that reconstructed by the traditional post-processing method. The average decrease in slope is 18.42%.

### 5.3. Cement Pavement Surface

The cement pavement surface to be measured has the area of 100 mm∗100 mm, as shown in Figure 11a. After the interference fringe pattern is projected onto it, the pavement fringe image is captured, as shown in Figure 11b.

Figure 11c,d includes the numerical surfaces reconstructed by the traditional post-processing method and the optimized post-processing method, respectively. The reconstructed 3D profile is drawn by different colors according to the magnitude of the depth values.

As shown in Figure 12, 1000 points on the selected lines (AB, CD, EF, GH, and IJ) are used to compare the slopes of the reconstructed surfaces acquired by the traditional post-processing method and the optimized post-processing method.

Table 3 shows the slope gradient of the reconstructed surface. The traditional post-processing method leads to the slant-reconstructed surface. The proposed post-processing method decreases the slope of the reconstructed surface by 15.29% compared with that reconstructed by the traditional post-processing method.

## 6. Results and Discussion

Previous studies focused on measuring the texture of workpieces with the diameter of <30 mm, and they used the traditional post-processing method to realize 3D reconstruction based on interference fringe. However, when measuring the pavement surfaces (diameter > 30 mm) with the traditional post-processing method, errors will cause reconstructed surface slant.

On condition that the slope is constant, the errors will increase when the length of the reconstructed surface increases, according to the characteristics of the similar triangle, such that the errors along axis Z increase with the extension of X, as shown in Figure 13. As a result, the problem of slant-reconstructed surface is quite essential to be solved, especially regarding the measurement of surfaces with larger area.

Experiments were completed on both the smooth surfaces of standard components and the coarse surfaces of pavement. Compared to the traditional post-processing method, for the case of smooth surface, the slope can be decreased by 69.00%; for the case of coarse surface, the slope can be decreased by 15.29%. 

The least squares method is used to estimate the slope of the whole reconstructed surface for statistical purposes. There is a difference between results gained from the smooth surface and the coarse surface for the following reason: as shown in Figure 14a, the coarse surface is uneven; the slopes of any two points on the reconstructed surface of pavement are quite different from any other ones. A–E are some points on the coarse surface being measured and F–H are some points on the smooth surface being measured. For example, the slope of AB is much different from that of CD or DE, resulting in the evaluated slope of the whole reconstructed surface being quite different from that of AB, CD, or DE. However, the slopes of any two points on the reconstructed surface of a smooth surface are almost the same as the evaluated slope: as shown in Figure 14b, the slopes of FG and GH approximate the evaluated slope of the smooth surface. The reason discussed above causes the results gained from the smooth surface but in a somewhat different manner from the coarse surface.

This study optimizes the post-processing method and solves the problem of slant-reconstructed surface by considering the influence of laser incident angle during the process of the post-processing method. Compared with the traditional post-processing method, the optimized post-processing method can reduce reconstruction errors by 74.51%. Further, 3D-PTRIF with the proposed post-processing method provides a better tool for engineers to calculate the international roughness index (IRI), texture depth (TD), and rutting depth index (RDI) of pavement, which could improve the accuracy of skid resistance measurement [26].

## 7. Conclusions

This paper elaborates on the optimized post-processing method of 3D-PTRIF, which considers the influence of unequal incident angles that contribute to the beam-divergence angle of the laser beam. The traditional post-processing method ignores the influence of the unequal laser incident angles, leading to erroneous measurement results.

In order to validate the effect of the optimized post-processing method, a standard component of texture depth, a piece of asphalt pavement, and a piece of cement pavement are measured, respectively. The standard component of texture depth represents a smooth surface and both pavement surfaces represent coarse surfaces. Moreover, it is demonstrated that the system of 3D pavement texture reconstruction based on interference fringe has a good effect on reconstructing the texture of both smooth surfaces and coarse surfaces, with accuracy reaching ±0.13 mm and resolution 0.07 mm. The result shows that 3D-PTRIF with the optimized post-processing method has higher accuracy than the traditional post-processing method and can reduce reconstruction errors by 74.51%. It solves the problem of slant-reconstructed surface. Compared to the traditional post-processing method, for the case of smooth surface, the slope can be decreased by 69.00%; for the case of coarse surface, the slope can be decreased by 15.29%. It is noted that 3D-PTRIF with the optimized post-processing method has a good effect on 3D reconstruction of pavement texture, which could improve the precision of skid resistance evaluation. There is a limitation of the current study that the diameter of the scanned area on the pavement surface being measured must be ≤180 mm. A future research direction could concentrate on the extension of the scanning area for 3D-PTRIF.

## Figures and Tables

**Figure 1 sensors-23-04660-f001:**
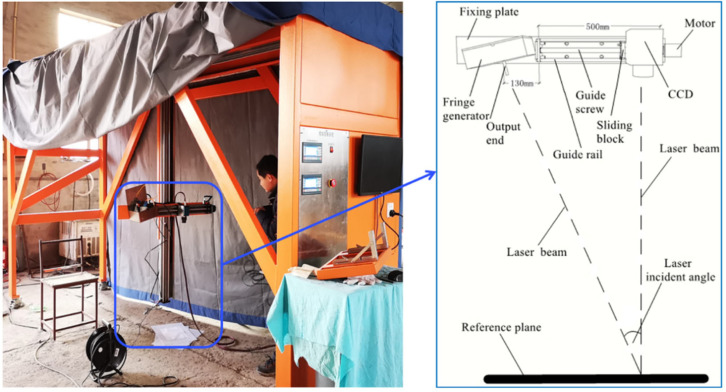
The system of 3D-PTRIF and its principles of the scanning system.

**Figure 2 sensors-23-04660-f002:**
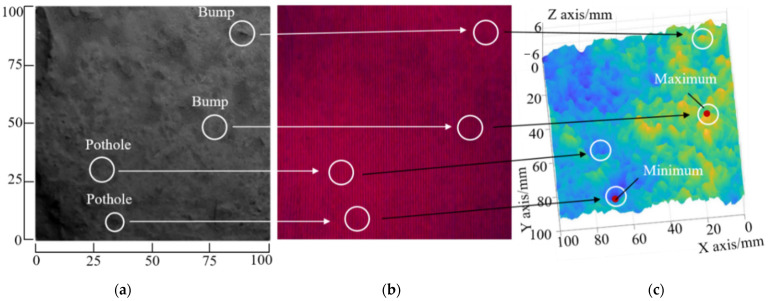
Tests on pavement surface: (**a**) the pavement surface being measured, (**b**) its fringe image, and (**c**) the corresponding 3D reconstruction map.

**Figure 3 sensors-23-04660-f003:**
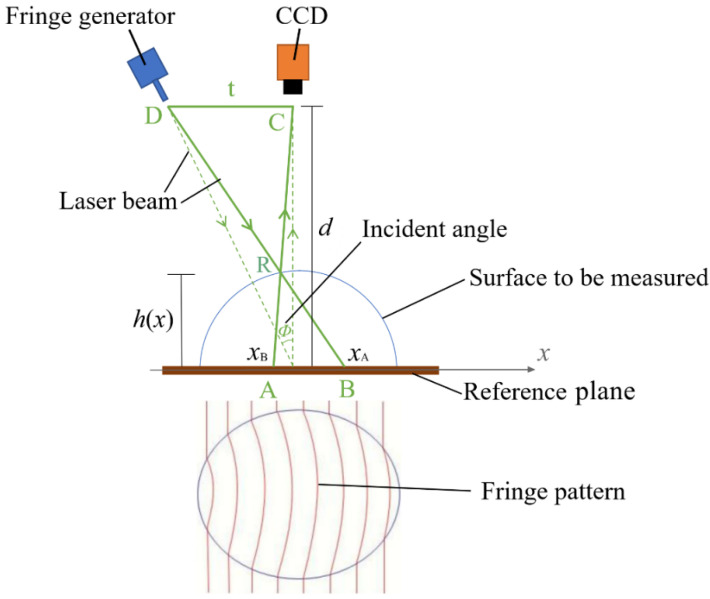
The principle of obtaining the texture depth data.

**Figure 4 sensors-23-04660-f004:**
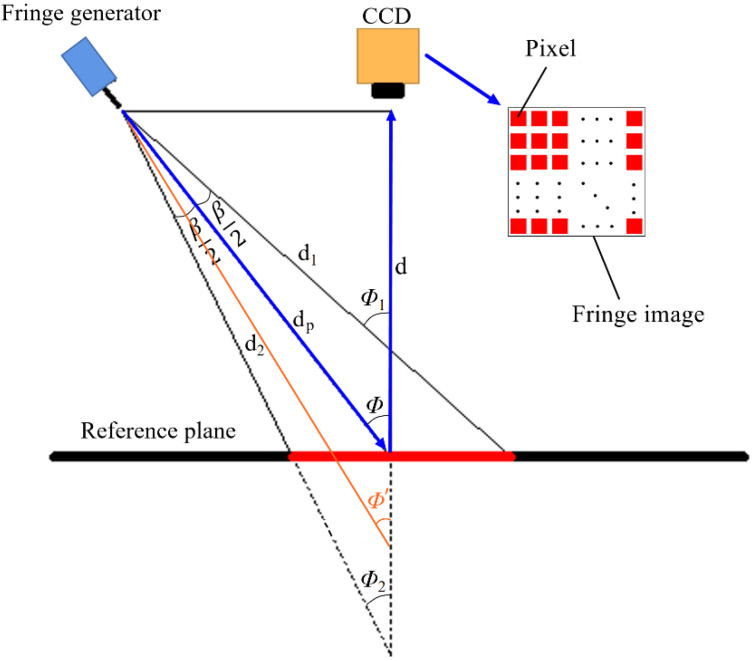
Unequal laser incident angles on the fringe pattern.

**Figure 5 sensors-23-04660-f005:**
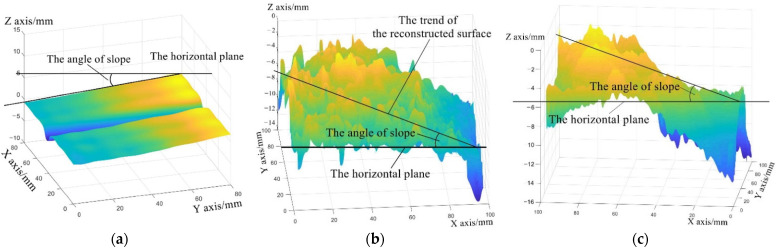
Examples of reconstructed surfaces that are deviated from the horizontal plane: (**a**) a reconstructed surface of the standard component, (**b**) a reconstructed surface of asphalt pavement, and (**c**) a reconstructed surface of cement pavement.

**Figure 6 sensors-23-04660-f006:**
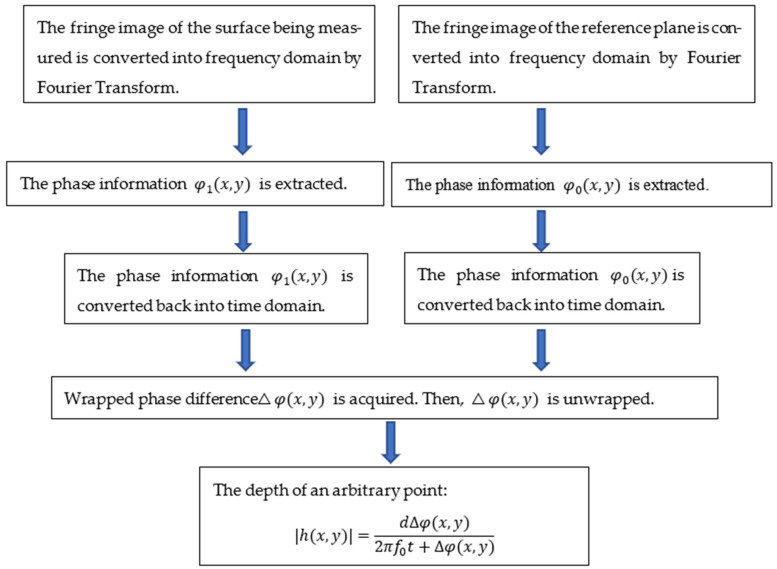
The traditional post-processing method of acquiring the texture depth.

**Figure 7 sensors-23-04660-f007:**
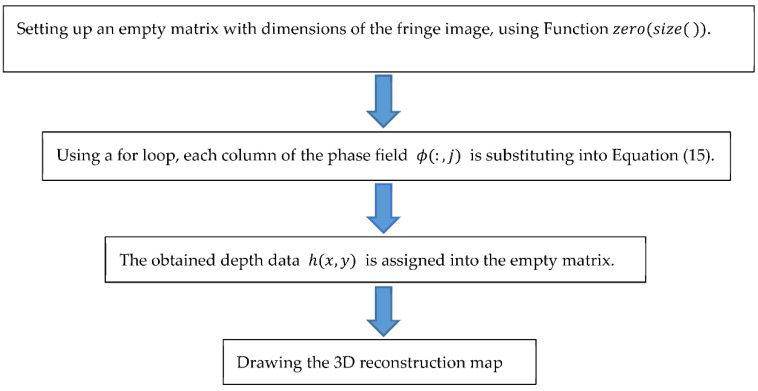
The optimized method of calculating the texture depth.

**Figure 8 sensors-23-04660-f008:**
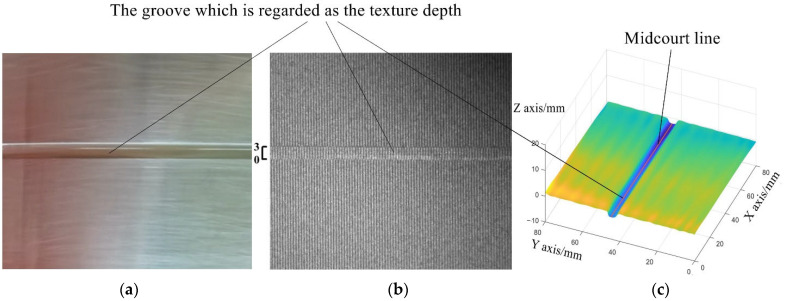
Texture depth: (**a**) the standard component of texture depth, (**b**) its fringe image, and (**c**) the selected points on the 3D reconstruction map.

**Figure 9 sensors-23-04660-f009:**
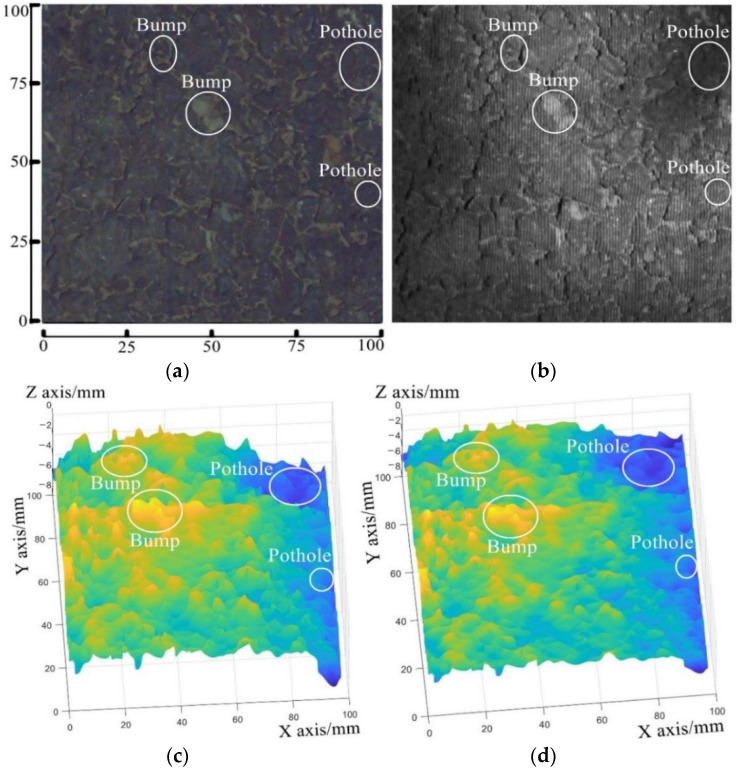
Tests on asphalt pavement surface: (**a**) the pavement surface being measured, (**b**) its fringe image, (**c**) the reconstructed surface generated by the traditional post-processing method, and (**d**) the reconstructed surface generated by the optimized post-processing method.

**Figure 10 sensors-23-04660-f010:**
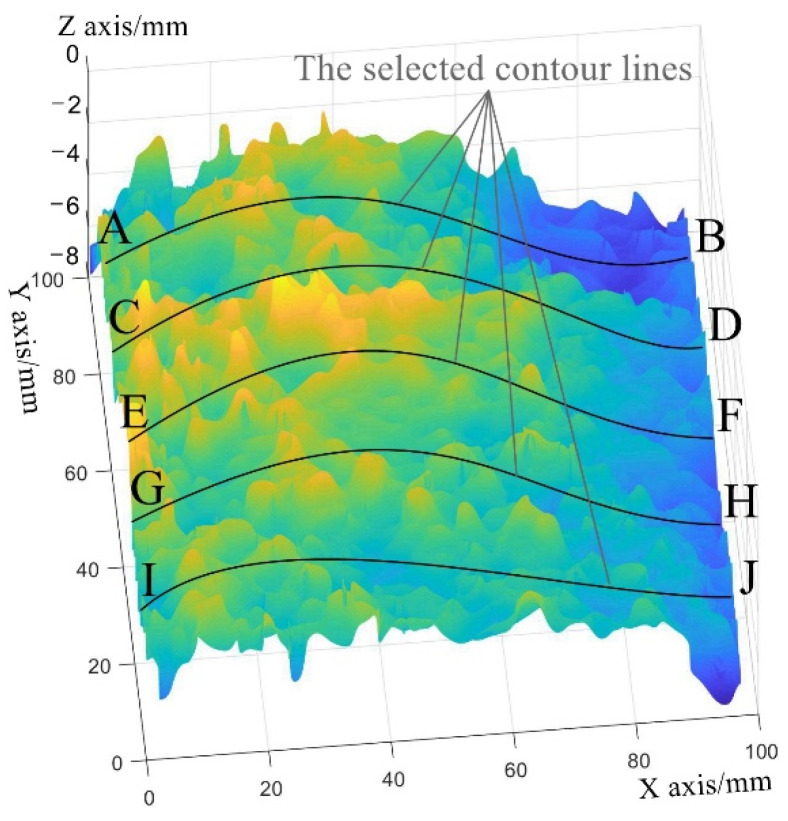
The selected contour lines that are used to evaluate the slope of the reconstructed surface.

**Figure 11 sensors-23-04660-f011:**
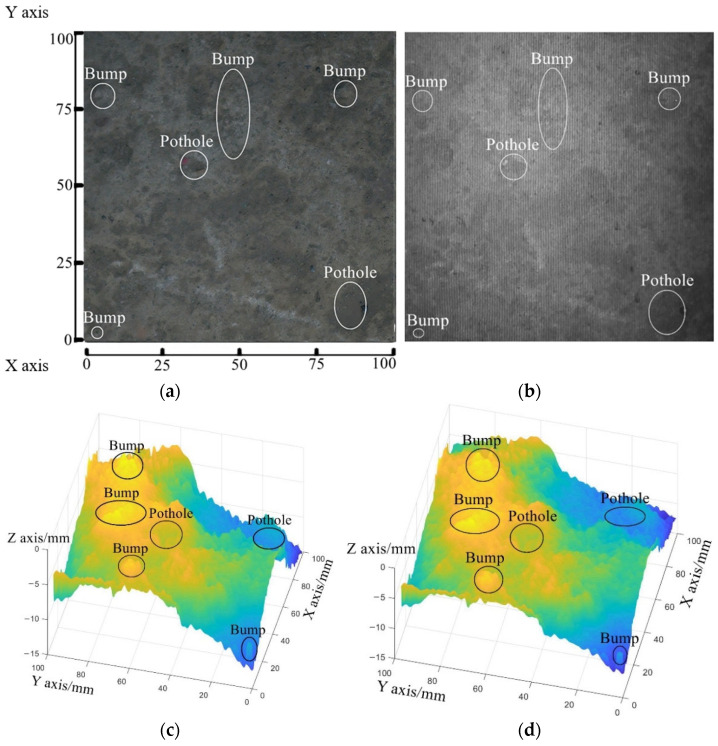
Tests on cement pavement surface: (**a**) the pavement surface being measured, (**b**) its fringe image, (**c**) the reconstructed surface generated by the traditional post-processing method, and (**d**) the reconstructed surface generated by the optimized post-processing method.

**Figure 12 sensors-23-04660-f012:**
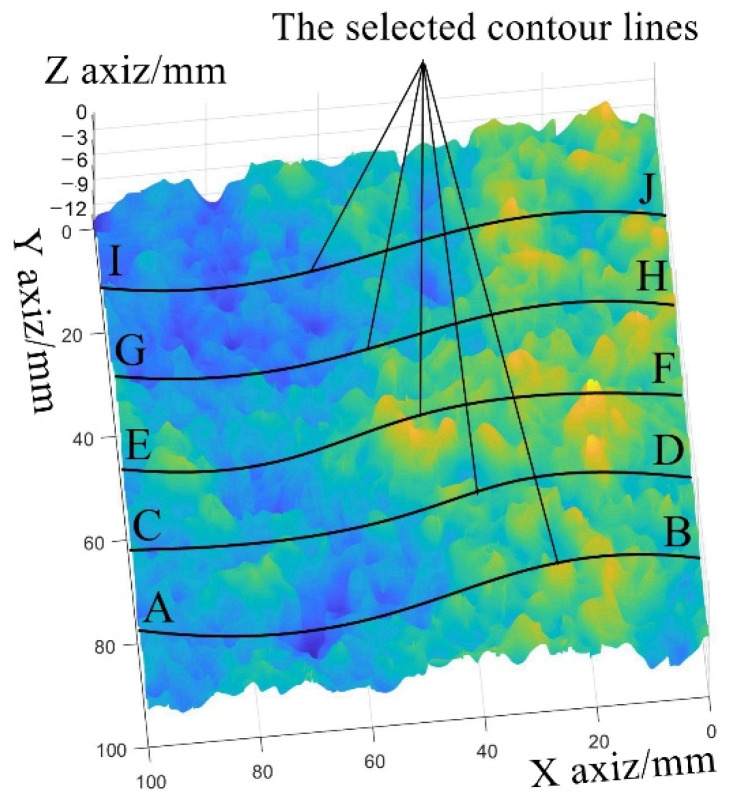
The selected contour lines on the reconstructed surface.

**Figure 13 sensors-23-04660-f013:**
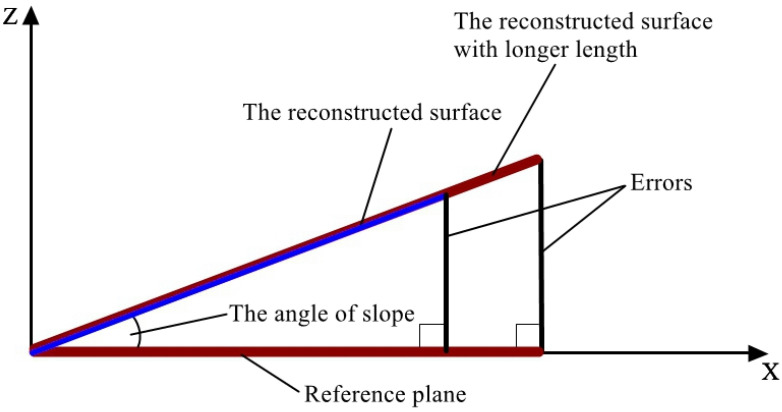
The characteristics of a similar triangle.

**Figure 14 sensors-23-04660-f014:**
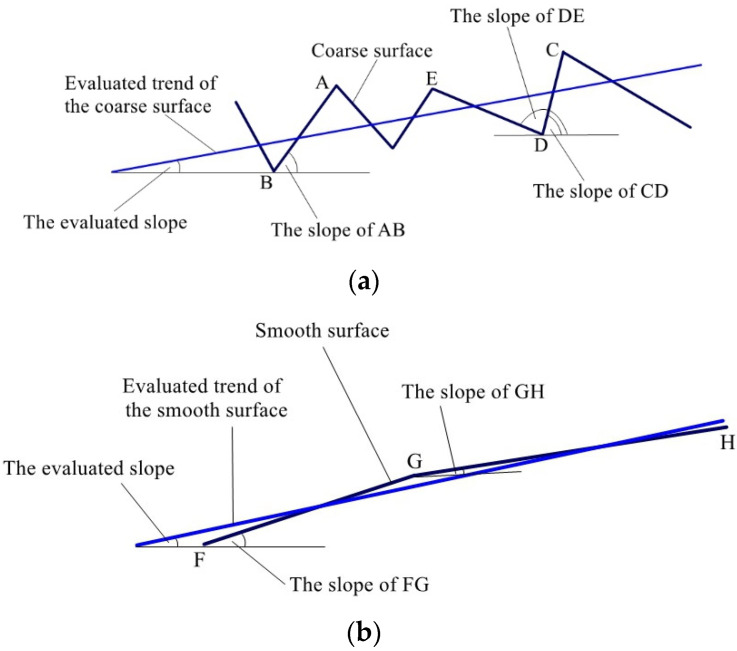
Evaluated slopes of smooth surface and coarse surface: (**a**) coarse surface and (**b**) smooth surface.

**Table 1 sensors-23-04660-t001:** The slope of the standard texture depth.

Method		Point Number				
		1	2	3	…	800
The traditional method	Depth (mm)	−1.4196	−1.4165	−1.4135	…	−2.1009
	Average Slope	−0.0016				
The optimized method	Depth (mm)	−1.9059	−1.9048	−1.9036	…	−2.0085
	Average Slope	−4.9598 × 10^−4^				
The decrease in the slope		69.00%				

**Table 2 sensors-23-04660-t002:** Comparing the slope.

Method		Selected Lines				
		AB	CD	EF	GH	IJ
Using the traditional method	Slope	−9.4531 × 10^−4^	−8.2163 × 10^−4^	−6.4581 × 10^−4^	−7.4953 × 10^−4^	−8.5340 × 10^−4^
Using the proposed method	Slope	−7.6484 × 10^−4^	−6.6913 × 10^−4^	−5.3011 × 10^−4^	−6.1198 × 10^−4^	−6.9828 × 10^−4^
Decrease in slope		19.09%	18.56%	17.92%	18.35%	18.18%
Average decrease in slope		18.42%				

**Table 3 sensors-23-04660-t003:** Comparing the slope.

Method		Selected Lines				
		AB	CD	EF	GH	IJ
Using the traditional method	Slope	−0.0014	−7.5465 × 10^−4^	−0.0012	−0.0017	−0.0021
Using the proposed method	Slope	−0.0012	−6.5223 × 10^−4^	−0.0010	−0.0014	−0.0018
Decrease in slope		14.29%	13.57%	16.67%	17.65%	14.29%
Average decrease in slope		15.29%				

## Data Availability

Some or all data, models, or code that support the findings of this study are available from the corresponding author upon reasonable request.

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
