# Peer review of "Improved 3D Pavement Texture Reconstruction Method Based on Interference Fringe via Optimizing the Post-Processing Method"

_sensors, 2023, doi:10.3390/s23104660_

Round 1
Reviewer 1 Report
The study found that traditional post-processing method of 3D Pavement Texture Reconstruction based on Interference Fringe(3D-PTRIF) ignore the divergence angle of the laser beam, resulting in different incident angles and insufficient accuracy in post-processing measurement data, which can lead to reconstructed surfaces that do not match reality. To address this problem, the authors proposed an optimized post-processing method that takes into account different incident angles, which can greatly improve measurement accuracy. The proposed method can definitely help the pavement engineers better understand the microstructure and profile of the test pavements. Therefore, the reviewer suggests to accept the paper after minor revision.
Comments:
Section 2.1:
-The authors claim that the system has a fast and efficient scanning mode, but how long does it typically take to scan a given area?
-Are there any limitations to the type of pavement surfaces that can be accurately measured with this system? For example, would highly reflective or dark surfaces be problematic?
-Interference fringe technology requires high measurement environment control, such as sensitivity to light, temperature, and vibration, thus requiring strict laboratory and environmental control. In addition, interference fringe technology may encounter issues when measuring objects with non-uniform surface color and reflectivity. Since interference fringe technology measures surface height differences through the phase difference of light, non-uniform surface color and reflectivity can alter the interference fringe pattern, affecting measurement accuracy. Do these conditions affect the actual performance of the equipment used in this article?
-Section5 Validation of the optimized post-processing method
The language in this section should be refined for greater clarity and concision, promoting fluency and precision of meaning.
-In the conclusion section, the practicality and application prospects of this method can be emphasized, and future research directions and significance can be proposed.
Reviewer 2 Report
Please, see the attached file.

Reviewer 3 Report
Improved 3D pavement texture reconstruction method based on interference fringe via optimizing the post-processing
Abstract
· The authors are recommended to display the basic methodology and findings in a quantitative manner.
introduction
· Literature review should be extended and some suggested studies can enhance the review.
· Road Materials and Pavement Design, Volume 23, 2022 - Issue 2, https://doi.org/10.1080/14680629.2020.1826347
· Research innovation should be stated at the end of previous studies
· The research objectives are not clear.
Methodology
· The authors should rethink about the display method of the laboratory part which should be more important in this study.
Results and Discussion:
· Page 4, line 110. Please write this part in form of paragraph
· The writing style is not attractive to the reader. Please rethink about it.
· Please compare the findings of this study with previous studies.
· How can this idea be useful for the designers engineers?
· How the skid resistance measurement be improved by using this technique?
Conclusion:
· The authors should give their findings in a quantitative manner.
· A recommendations to the practitioners should be added. Also a limitation of the current study should be highlighted.
References:
The format of references is not compatible with the journal format.
Reviewer 4 Report
I suggest that the authors must consider the following corrections:
1. The author should explain clearly in the Abstract and in Introduction, what is the novelty of the proposed method and what is the added value in this article.
2. The author should check typing errors throughout the manuscript. Improve the quality of editing the text and equations. Use the points at the end of the equations. English style should also be improved.
3. I think the authors need to emphasize more clearly the contribution of the manuscript from a scientific point of view.
4. Author, please write the references in the same mode as required by the journal and put the DOI number where possible. I am convinced that it is useful for the manuscript if it will be included in the References section following papers with the same topics or using similar procedures, ex.: Mechanics of Elastic Composites, Chapman & Hall/ CRC Press, U.S.A, 708 pp., (2003); Reconstruction of 3D Pavement Texture on Handling Dropouts and Spikes Using Multiple Data Processing Methods, Sensors 2019, 19(2), 278; https://doi.org/10.3390/s19020278.
Round 2
Reviewer 2 Report
Please, download the attached file.

Reviewer 3 Report
The authors addressed the reviewer comments in a reasonable way.
Author Response
We are so grateful for the valuable comments given by Reviewer 3.